# Brain Iron and Mental Health Symptoms in Youth with and without Prenatal Alcohol Exposure

**DOI:** 10.3390/nu14112213

**Published:** 2022-05-26

**Authors:** Daphne Nakhid, Carly A. McMorris, Hongfu Sun, Ben Gibbard, Christina Tortorelli, Catherine Lebel

**Affiliations:** 1Department of Neuroscience, University of Calgary, Calgary, AB T2N 1N4, Canada; daphne.nakhid@ucalgary.ca; 2Alberta Children’s Hospital Research Institute (ACHRI), University of Calgary, Calgary, AB T2N 1N4, Canada; camcmorr@ucalgary.ca (C.A.M.); ben.gibbard@albertahealthservices.ca (B.G.); 3Hotchkiss Brain Institute, University of Calgary, Calgary, AB T2N 1N4, Canada; 4School and Applied Child Psychology, Werklund School of Education, University of Calgary, Calgary, AB T2N 1N4, Canada; 5Department of Pediatrics, University of Calgary, Calgary, AB T2N 1N4, Canada; 6School of Information Technology and Electrical Engineering, University of Queensland, Brisbane 4072, Australia; hongfu.sun@uq.edu.au; 7Department of Child Studies and Social Work, Mount Royal University, Calgary, AB T3E 6K6, Canada; ctortorelli@mtroyal.ca; 8Department of Radiology, University of Calgary, Calgary, AB T2N 1N4, Canada

**Keywords:** prenatal alcohol exposure, FASD, brain, iron, QSM, MRI, mental health, internalizing symptoms, externalizing symptoms, youth

## Abstract

Prenatal alcohol exposure (PAE) negatively affects brain development and increases the risk of poor mental health. We investigated if brain volumes or magnetic susceptibility, an indirect measure of brain iron, were associated with internalizing or externalizing symptoms in youth with and without PAE. T1-weighted and quantitative susceptibility mapping (QSM) MRI scans were collected for 19 PAE and 40 unexposed participants aged 7.5–15 years. Magnetic susceptibility and volume of basal ganglia and limbic structures were extracted using FreeSurfer. Internalizing and Externalizing Problems were assessed using the Behavioural Assessment System for Children (BASC-2-PRS). Susceptibility in the nucleus accumbens was negatively associated with Internalizing Problems, while amygdala susceptibility was positively associated with Internalizing Problems across groups. PAE moderated the relationship between thalamus susceptibility and internalizing symptoms as well as the relationship between putamen susceptibility and externalizing symptoms. Brain volume was not related to internalizing or externalizing symptoms. These findings highlight that brain iron is related to internalizing and externalizing symptoms differently in some brain regions for youth with and without PAE. Atypical iron levels (high or low) may indicate mental health issues across individuals, and iron in the thalamus may be particularly important for behavior in individuals with PAE.

## 1. Introduction

Prenatal alcohol exposure (PAE) can affect fetal development, resulting in altered neuroanatomy, neurophysiology, and increased risk of poor mental health outcomes. Fetal alcohol spectrum disorder (FASD) is the neurodevelopmental disability associated with PAE and the most common preventable cause of developmental disabilities in children [1,2,3]. PAE is linked to aberrant structural and functional brain development, including globally reduced brain volume, reduced white and gray matter volume (see reviews [4,5]), altered cortical thickness [6], altered white matter connectivity [7,8], and abnormal brain function [9].

Approximately 90% of individuals with FASD have mental health problems [10]. Evidence of mental health problems in infants with PAE can be observed as early as 2 years of age [11] and continues to increase through childhood and adolescence, persisting into adulthood [10]. Children with PAE experience more mental health problems than their ability-matched peers [12]. Mental health problems in children can be thought of as internalizing (negative behaviors directed inwards, such as anxiety or depression) or externalizing (negative behaviors directed outwards, such as aggression and hyperactivity). Internalizing disorders frequently co-occur with FASD and are 4 and 11 times more common than in the general population, respectively [13]. Similarly, externalizing disorders such as attention-deficit/hyperactivity disorder (ADHD) and oppositional defiant disorder are 10 and 13 times more common in individuals with PAE than in the general population, respectively [13]. Unexposed youth and adults with internalizing and externalizing disorders tend to show volumetric abnormalities of limbic and basal ganglia structures [14,15,16,17,18,19,20,21,22]. Youth with PAE have similar volumetric abnormalities [23,24,25]; however, little research has investigated the neural underpinnings of internalizing or externalizing symptoms in individuals with PAE.

In the general population, brain iron, and in particular, iron deficiency anemia, has been linked to developmental disabilities and mental health problems in youth [26] and adults [27]. Early brain development requires iron for myelination [28,29], monoamine synthesis, and neurotransmitter metabolism [29,30]. In the first six months of life, infants depend on fetal iron stores accumulated during the third trimester of pregnancy [31,32]. Infants and children who have an iron deficiency show increased cognitive and affective problems (e.g., less social smiling and looking) [33,34]. Altered developmental and mental health outcomes associated with iron deficiency persist for years even after treatment [35], highlighting the importance of sufficient iron availability throughout development.

Literature investigating brain iron in individuals with PAE is very limited. Prenatal maternal binge drinking increases the risk of iron deficiency anemia in infants at 12 months of age [31]. In animal models, PAE alters brain iron bioavailability [36], lowers fetal brain iron [37], and behavioral deficits persist even after iron levels normalize [38]. Blood iron deficiency in infancy is related to emotional withdrawal, which predicts affective problems in childhood [39]. Of note, the relation between blood iron and brain iron is not well understood, and blood iron does not always correlate with brain iron, especially in children with neurodevelopmental disabilities [40]. In other neurodevelopmental disabilities, preschoolers with autism have lower brain iron in the caudate nucleus [41], and children with ADHD have lower brain iron in the caudate, putamen, and thalamus compared to controls [40,41,42]. Although brain iron is associated with poor developmental and mental health outcomes often experienced by individuals with PAE, no studies to date have looked at brain iron in youth with PAE and how it is associated with internalizing and externalizing issues.

Transverse relaxation rates (*R*2*) and susceptibility-weighted imaging (SWI) have both been validated as correlated to iron concentration [43,44,45,46], though they are indirectly related to brain iron and lack specificity [47,48]. Quantitative susceptibility mapping (QSM) measures tissue susceptibility, an intrinsic property that determines how tissue will interact with a magnetic field and provides a more reliable, reproducible, and valid measure of brain iron. Iron is paramagnetic (positive susceptibility) and has been validated post mortem as the primary susceptibility source in subcortical gray matter [49,50].

Here, we examined the association between brain iron and internalizing or externalizing symptoms in youth and determined how the group (exposed versus unexposed) moderates this association. Consistent with previous literature, we focused on key brain regions: the basal ganglia (nucleus accumbens, caudate, putamen, and pallidum), a primary site of iron accumulation [50,51], as well as limbic structures (thalamus, hippocampus, and amygdala) that have been implicated in both PAE and mental health problems [5,14,15].

Based on previous research on brain iron in children with externalizing symptoms [40,41,42] and adults with internalizing symptoms [52,53], it was expected that across diagnostic groups, a lower susceptibility of the caudate and putamen would be associated with more internalizing and externalizing symptoms. Further, a group–susceptibility interaction was anticipated, such that lower susceptibility in the thalamus would only be associated with more internalizing and externalizing symptoms in the PAE group. Consistent with Krueger and colleagues, who found a trend-level association between caudate volume and internalizing symptoms in children with PAE [54], we hypothesized that the PAE group would have a lower caudate volume and that this would be associated with more internalizing symptoms.

## 2. Materials and Methods

### 2.1. Participants

*PAE group.* Nineteen youth with PAE (M = 11.18 years, SD = 2.16; 10 boys (B)/9 girls (G); Table 1) aged 7.5–15 years were recruited through posters, social media, word of mouth, caregiver support groups, early intervention services, and the Cumulative Risk Diagnostic Clinic in Calgary, Alberta (Canada). PAE was confirmed through child welfare files, medical, police, and social work records (when available), as well as semi-structured interviews with caregivers and/or birth families. Thirteen participants with PAE had confirmed alcohol exposure, meeting the Canadian diagnostic guidelines for FASD (≥7 drinks/week or ≥2 binge episodes of at least 4 drinks at some point in pregnancy), and six participants with PAE had confirmed exposure to an unknown amount. Seven participants had a diagnosis of FASD, and one participant was classified as at risk of developing FASD. Some participants with PAE had a prior diagnosis of ADHD (*n* = 8), learning disability (*n* = 1), anxiety (*n* = 1), attachment disorder (*n* = 2), borderline intellectual functioning (*n* = 1), mild mental disability (*n* = 1), and oppositional defiant disorder (*n* = 1); three participants with PAE had more than one diagnosis. Ten participants were taking medication (seven for ADHD symptoms, three for other symptoms) at the time of their scan. No participants had MRI contraindications. Gender was parent/caregiver reported.

*Unexposed group.* Forty unexposed youth (M = 11.18 years, SD = 2.27; 23B/17G; Table 1) aged 7.5–15 years were recruited through posters, social media, and word of mouth. These participants had confirmed absence of PAE based on retrospective questionnaires completed by a caregiver at the time of the MRI scan. These participants had no prior or suspected diagnosis of a neurodevelopmental disability or mental health issues, and none of these participants were taking medication related to developmental and mental health difficulties. Participants had no MRI contraindications, and gender was parent/caregiver reported.

### 2.2. MRI Acquisition

Participants completed an MRI scan on a 3T GE MR750w MRI scanner at the Alberta Children’s Hospital using a 32-channel head coil. The imaging protocol included a T1-weighted anatomical scan acquired using a 3D FSPGR sequence with TI = 600 ms, TR = 8.2 ms, 0.8 mm isotropic resolution, and total scan time of 5:38 min. QSM images were acquired using a 3-dimensional spoiled gradient recalled sequence with the acquisition parameters: repetition time (TR) = 56.8 ms, Echo Time (TE) = 4.5 ms to 52.2 ms for a total of 10 echoes with 5.3 ms spacing between each echo, field of view = 240 × 240 × 182.4 mm^3^, acquired voxel size = 0.94 × 1.17 × 1.90 mm^3^, reconstructed voxel size = 0.94 × 0.94 × 1.90 mm^3^, flip angle = 10°, acceleration factor = 2.5, and total scan time of 5:14 min.

### 2.3. T1-Weighted Image Processing

FreeSurfer version 6.0 (Boston, MA, USA) automated recon-all pipeline was used for processing, editing, and segmenting the T1-weighted anatomical images [55,56]. The recon-all pipeline registers participants’ brains to a template brain, performs skull stripping, image registration, intensity normalization, segmentation, and performs volume calculations. Segmentations were manually checked to ensure proper segmentation of the outer pial and white matter border, and manual editing was performed as necessary. We focused on the basal ganglia, a key site of iron accumulation [50,51], and limbic structures that are implicated in PAE and mental health [5,14,15]. The segmentations from FreeSurfer version 6.0 (Boston, MA, USA) were used to extract the bilateral caudate, putamen, pallidum, thalamus, amygdala, hippocampus, and nucleus accumbens (Figure 1).

### 2.4. QSM Reconstruction

The first five echoes (TE from 4.5 ms to 25.7 ms) were used for QSM reconstruction. The raw phase was unwrapped using a best-path method [58], RESHARP [59] was used for background field removal, and iLSQR [60] was used for dipole inversion. The QSM processing pipeline was implemented in MATLAB version R2018a (Natick, MA, USA) [61] source code is available on GitHub: https://github.com/sunhongfu/QSM (accessed on 14 June 2019). Mean susceptibility of the whole brain was used as a point of reference since there is no susceptibility bias during analysis, whether referencing the whole brain or a specific region [62,63,64]. Segmentations from FreeSurfer version 6.0 (Boston, MA, USA) were transformed to QSM space using mri_robust_register [65] and were used to extract the susceptibility values of the caudate, putamen, pallidum, thalamus, amygdala, hippocampus, and nucleus accumbens bilaterally.

### 2.5. Behavioural Measures

Caregivers completed the Behavioural Assessment System for Children, Second Edition–Parent Rating Scale (BASC-2-PRS) [66] to measure Internalizing Problems (Anxiety, Depression, Somatization) and Externalizing Problems (Aggression, Conduct, and Hyperactivity). The BASC-2 provides *T* scores that are age- and gender-normed; higher scores indicate worse symptoms. Scores <60 are considered developmentally appropriate, scores of 60–69 are considered at-risk of developing a disorder, and scores ≥70 are considered clinically maladaptive behavior indicating that a child may meet diagnostic criteria for a disorder. The Internalizing Problems composite score includes Anxiety, Depression, and Somatization subscales, while the Externalizing Problems composite score includes Aggression, Conduct Problems, and Hyperactivity subscales. Composite *T* scores are the sum and norm of the subscale *T* scores and provide a broader measure of behavior.

### 2.6. Statistical Analysis

R Statistics version 4.0.0 (Boston, MA, USA) [67] was used for all analyses. To determine if groups differed by age and gender, a chi-square analysis (gender) and independent sample *t*-test (age) were conducted. To determine if diagnostic groups differed on Internalizing and Externalizing Problems, a two-sample Wilcoxon rank test was used. ANCOVAs for each brain region were conducted to determine the association between group and susceptibility or volume, with age and gender as covariates. To determine if there was a main effect of group (PAE and unexposed), brain measures (susceptibility/brain iron or volume), or a group–brain interaction, one-way ANCOVAs for each brain region were conducted with age and gender as covariates. If the group–brain interaction was not significant, it was removed from the model. For structures where susceptibility or volume was significantly associated with behavior, we also tested associations using a one-way ANCOVA with Internalizing Problems (Depression, Anxiety, Somatization) or Externalizing Problems (Hyperactivity, Conduct Problems, Aggression) subscales. Results are reported both uncorrected (*p* < 0.05) and corrected for multiple comparisons using the Benjamini–Hochberg false-discovery rate (FDR) at *q* < 0.05.

## 3. Results

### 3.1. Demographics

There were no group differences in gender (X^2^(1, *n* = 59) = 0.124, *p* = 0.725) or age (*t* (37.057) = −0.005, *p* = 0.996); see Table 1.

### 3.2. Diagnostic Group Differences in Susceptibility and Volume

The PAE group had lower susceptibility in the hippocampus than the unexposed group (F = 4.57, *p* = 0.037; Table 2), but this did not survive FDR correction (*q* = 0.259). No other regions had significant group differences in susceptibility (Table 2).

The PAE group had smaller volumes than the unexposed group in the caudate (F = 16.7, *p* < 0.001, *q* < 0.001), putamen (F = 5.92, *p* = 0.018, *q* = 0.042), and pallidum (F = 18.2, *p* < 0.001, *q* < 0.001), all of which survived FDR corrections (Table 3).

### 3.3. Behavioural Outcomes

The median Externalizing T score in the PAE group (68; IQR = 10) was significantly higher than in the unexposed group (50; IQR = 9; *p* < 0.001); see Table 4. The median Aggression T score in PAE group (60; IQR = 11) was significantly higher than in the unexposed group (50; IQR = 8; *p* < 0.001). The median Conduct T score in the PAE group (60; IQR = 16) was significantly higher than in the unexposed group (51; IQR = 14; *p* = 0.002). The median Hyperactivity T score in the PAE group (69; IQR = 14) was significantly higher than in the unexposed group (50; IQR = 10; *p* < 0.001). There were no significant group differences in internalizing scores.

### 3.4. Magnetic Susceptibility and Behavioural Outcomes

Across groups, susceptibility of the nucleus accumbens was negatively associated with Internalizing *T* scores (F(1, 54) = 7.020, *p* = 0.011, *q* = 0.056; Table 5), as well as Depression (F(1, 54) = 5.447, *p* = 0.023, *q* = 0.069) and Anxiety (F(1, 54) = 12.455, *p* < 0.001, *q* = 0.003) subscales (Figure 2). Amygdala susceptibility was positively associated with Internalizing *T* scores (F(1, 54) = 4.168, *p* = 0.046, *q* = 0.107) and the Anxiety subscale (F(1, 54) = 5.009, *p* = 0.029, *q* = 0.029; Figure 3). There was a group–susceptibility interaction for the Somatization subscale (F(1, 53)= 4.121, *p* = 0.047, *q* = 0.141), such that there was a more positive association between amygdala susceptibility and Somatization symptoms in the PAE group than in unexposed participants (Figure 3).

There was a group–susceptibility interaction for the thalamus for Internalizing *T* scores (F(1, 53) = 6.157, *p* = 0.016, *q* = 0.056; Table 5), Anxiety (F(1, 53) = 9.305, *p* = 0.004, *q* = 0.006), and Depression (F(1, 53) = 4.043, *p* = 0.049, *q* = 0.074), such that scores were more negatively correlated with susceptibility for the PAE group than controls (Figure 4). Only the relationship for the Anxiety subscale survived multiple comparison correction.

There was a group–susceptibility interaction for the putamen for Externalizing Problems (F(1, 53) = 4.497, *p* = 0.039, *q* = 0.273; Table 5), with the PAE group showing more Externalizing Problems associated with higher susceptibility and the unexposed group having a slightly negative correlation (Figure 5). None of the externalizing subscales was significantly related to putamen susceptibility.

### 3.5. Brain Volume and Behavioural Outcomes

There were no brain volume or group-volume interactions associated with Internalizing or Externalizing Problems (see Table 6).

## 4. Discussion

For the first time, we showed that brain iron is associated with internalizing and externalizing symptoms in youth and, in some cases, was moderated by PAE. Across groups, less brain iron (lower susceptibility) in the nucleus accumbens was associated with more internalizing symptoms, while in the amygdala, higher brain iron (greater susceptibility) was associated with more internalizing symptoms. This suggests that the effects of brain iron on mental health symptoms differ regionally. Our findings in the thalamus and putamen demonstrate that PAE not only alters brain iron but also disrupts the brain–behavior relationship. No associations between brain volumes and mental health symptoms were found, suggesting that more specific measures such as susceptibility may be helpful in determining what may be contributing to mental health problems in this population.

As expected, the PAE group had significantly more externalizing symptoms than unexposed controls. Externalizing symptoms are well-recognized in individuals with PAE; ADHD (an externalizing disorder) occurs 10 times more often in people with PAE than those unexposed, and it is the most common co-occurring disorder with FASD [13]. Internalizing symptoms did not differ significantly between groups. Many studies report that externalizing symptoms are more affected by PAE than internalizing symptoms at this age [12]. However, internalizing symptoms can be hard to measure with parent reports since internalizing symptoms are not as easily observed, which may contribute to an under-reporting of internalizing symptoms across one or both groups.

Iron is an essential nutrient for proper brain development and behavior. Alcohol consumption during pregnancy can affect fetal iron stores [37], disrupting postnatal brain development via myelination, neurotransmitter synthesis and metabolism, or ATP synthesis. While PAE may affect iron in white matter, QSM is best suited to detecting iron in gray matter, and therefore we focused on gray matter structures. QSM can distinguish between paramagnetic and diamagnetic material [68,69]; however, it cannot identify the specific components contributing to susceptibility. In subcortical structures, myelin and calcium contribute diamagnetic (negative) susceptibility [69,70] and heme iron, FeS clusters, iron isotopes, transferrin, and ferritin contribute paramagnetic (positive) susceptibility. More than 97% of iron in cells (and up to 90% of the iron in the brain) is stored as ferritin [70,71], and post mortem studies show that ferritin is the primary susceptibility source in subcortical structures [49,50]. Increased ferritin iron stores may reflect sufficient brain iron available for use, stored iron that is unavailable, or an overload of brain iron. Raghunathan and colleagues [72] propose that PAE alters brain vasculature, resulting in decreased brain vessel diameter, which may decrease heme iron. However, heme iron is not expected to contribute significantly to susceptibility [73] in subcortical brain structures. Therefore, the present study findings likely reflect primarily contributions from ferritin in the subcortical brain regions.

The nucleus accumbens plays an important role in motivation and emotional processing [74] and is implicated in a number of psychological disorders, including depression and anxiety [75]. Based on previous research, we had no hypothesis about brain iron in the nucleus accumbens and internalizing or externalizing symptoms, but found it was associated with internalizing symptoms, driven mostly by anxiety symptoms. The nucleus accumbens is part of the mesolimbic dopaminergic pathway, so low iron may impact behavior through dopamine metabolism. Iron-deficient rats show decreased dopamine transporter (DAT) in the striatum and nucleus accumbens, which can result in elevated dopamine, leading to desensitization of the DAT [76]. Similarly, Youdim [77] found decreased dopamine 2 (D2) receptors in the nucleus accumbens of iron-deficient rats; D2 receptors reduce acetylcholine release [78], the main neurotransmitter of the parasympathetic nervous system. While this evidence suggests that iron in the nucleus accumbens affects dopamine synthesis and metabolism, other neurotransmitters such as GABA, glutamate, and serotonin may also be affected.

The amygdala plays an important role in limbic circuits and emotional responses, including fear and anxiety [79]. Amygdala iron was positively related to Internalizing Problems and Anxiety symptoms in youth with and without PAE. We found a significant group–susceptibility interaction effect in the amygdala for Somatization symptoms. Post hoc testing showed that the PAE group had a more positive Somatization–susceptibility relationship, and the unexposed group had a slightly negative association, though neither of the relationships within groups was significant. Inhibitory circuits in the amygdala, which are highly dependent on GABA, are important for promoting and suppressing anxiety-like behavior [80]. The amygdala tends to have stable, low levels of brain iron across the lifespan [81]. It is possible that small perturbations in brain iron have a large effect on GABA metabolism and, therefore, on anxiety-like behavior in this brain region.

The thalamus is involved in limbic, cognitive, and sensorimotor circuits and acts primarily as a relay station [82]. We found a group–susceptibility interaction effect in the thalamus, where thalamus susceptibility and internalizing symptoms (Internalizing Problems, Depression and Anxiety subscales) were more negatively associated in the PAE group than in the unexposed group. The post hoc within-group association was only significant in the PAE group for Internalizing Problems and Anxiety symptoms. The lack of relationships with externalizing symptoms is surprising given that prior studies report lower thalamus iron in youth with ADHD [40,42], an externalizing disorder that often co-occurs with PAE. That being said, participants in this study with ADHD were almost all taking psychostimulant medication, which is thought to normalize brain iron in the thalamus [83]. Previous research in adults shows increased brain iron in the thalamus and putamen is related to depression severity [52,53]. We saw evidence of this trend in the unexposed group that had increasing internalizing symptoms with higher brain iron, while the opposite relationship was observed in youth with PAE. The thalamus is rich in dopamine [84], and behavioral outcomes related to brain iron in the thalamus are likely related to dopamine metabolism.

The putamen is part of the striatum involved in reward, cognition, language, and addiction [85]. We hypothesized that lower susceptibility in the putamen would be associated with more internalizing and externalizing symptoms. Instead, we found a group–susceptibility interaction effect such that the susceptibility in the putamen was more positively associated with Externalizing Problems in the PAE group than in the unexposed group. However, neither of the post hoc within-group relationships were significant. Previous research suggests that brain iron in the putamen is lower in children with ADHD [40]; however, in our sample, more brain iron in the putamen was associated with more externalizing symptoms in the PAE group. Previous research has found that iron in the putamen is related to depression severity in adults [52], but we did not find this in our sample. The putamen undergoes prolonged development [86], and it is possible that the effects of susceptibility on the putamen and behavior would be better captured in older adolescence or young adulthood. Although the mechanism of how brain iron is related to internalizing and externalizing symptoms is not well understood, it is likely due to dopamine, serotonin, and norepinephrine metabolism [87]. It is important to note that the concentration of brain iron required for healthy brain development differs by region [62]. Therefore, while the caudate and pallidum have a high deposition of brain iron throughout development, other structures with lower brain iron, such as the hippocampus and amygdala, may be more sensitive to small perturbations in brain iron.

We found lower susceptibility in the hippocampus for the PAE group compared to the unexposed participants, but this did not survive correction for multiple comparisons. The hippocampus is highly vulnerable to iron status during gestation [88], and while we did not find that hippocampus susceptibility was related to mental health symptoms, it may be associated with cognitive and learning difficulties that are common in individuals with PAE.

In contrast to the QSM results, there were no associations between brain volume and behavior. This differs from the research on mental health and brain structure in unexposed participants that has found associations between caudate volume, putamen development, amygdala, hippocampus, and thalamus abnormalities and internalizing and externalizing disorders [14,15,16,17,18,19,20,21,22]. Only one study of children with PAE found a trend-level association between caudate volume and internalizing symptoms [54]. Another study, in a sample overlapping with our study, found no associations between brain volumes and behavior [89]. One explanation is that volume reductions in children with PAE may disrupt the etiology of mental health problems, obscuring the relationship that may be seen in unexposed individuals. However, further research to clarify these relationships is needed.

Nutritional interventions, such as choline supplementation, show promise for promoting neurodevelopment in infants and children with PAE [90,91,92]. This opens the question of whether iron supplementation may be a viable intervention for individuals with PAE. Animal research has shown that iron supplementation can mitigate some of the effects of alcohol exposure during development [38,93,94]. Future studies with prospective and large longitudinal samples are required to fully understand the impact of PAE on brain iron to assess iron supplementation as a potential nutritional intervention.

### Limitations

This is the first study to examine brain iron and mental health in children and youth with PAE; however, some limitations in the study design and methodology exist. Gender was parent/caregiver reported, and biological sex was not collected. Future studies should collect both gender and sex when possible, as both variables provide distinct contributions to understanding the complex association between brain structure and mental health in children with PAE. QSM provides a novel way to study brain iron in children with PAE, though it cannot distinguish the components contributing to susceptibility. Given our small sample, this study had limited power. Information about PAE was collected as rigorously as possible; however, given the retrospective nature of the study, details around exposure amounts, trimester, and frequency were not always available. Youth with PAE are at a greater risk of experiencing other prenatal exposures (e.g., tobacco, cocaine, marijuana) [95,96] and postnatal adversities (e.g., neglect, deprivation, physical abuse, and caregiver changes) [95,97]; however, these factors were not considered in our analyses. Considering that prenatal exposures and postnatal adversities negatively impact brain development [89,95], these should be controlled in future studies. Lastly, mental health symptoms were measured using a caregiver report, which is less reliable when measuring internalizing symptoms [98] and may be underestimating internalizing symptoms. Given the cognitive and emotional challenges youth with PAE face, a caregiver report was chosen to be the best method for assessment collection; however, future studies should use both parent and self-report outcome measures when possible.

## 5. Conclusions

Expanding on animal research, we show that brain iron is associated with internalizing symptoms, primarily anxiety, in multiple brain regions in youth with and without PAE. Group–brain interactions in internalizing and externalizing symptoms in some brain regions suggest that brain iron may impact behavior differently in youth with and without PAE. Our lack of findings linking brain volume to internalizing or externalizing symptoms suggests a need for more specific brain measures, such as QSM, to better understand the neurological underpinnings of how PAE affects the brain and behavior.

## Figures and Tables

**Figure 1 nutrients-14-02213-f001:**
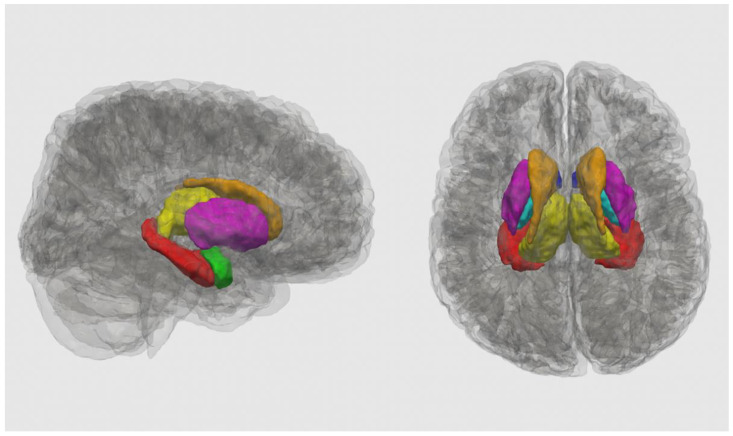
Regions of interest are shown in an unexposed control participant (girl, age 9 y) in a sagittal (**left**) and axial (**right**) 3D mesh reconstruction using ITK-Snap version 3.8.0 (USA) [57]: nucleus accumbens (dark blue), caudate (copper), putamen (pink), pallidum (cyan), thalamus (yellow), hippocampus (red), and amygdala (lime green).

**Figure 2 nutrients-14-02213-f002:**
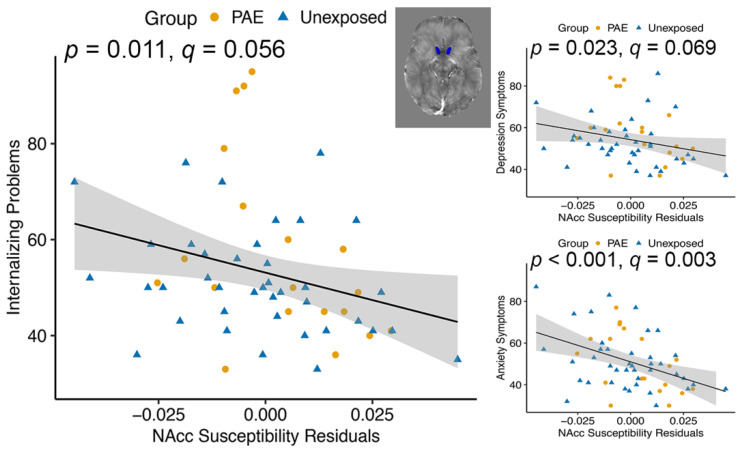
Nucleus accumbens (NAcc) magnetic susceptibility was negatively associated with BASC-2-PRS *T* scores Internalizing Problems, Depression, and Anxiety symptoms in youth with and without PAE. The *p* values indicate the main effect of susceptibility on Internalizing Problems, Depression, and Anxiety, and susceptibility values are corrected for age and gender.

**Figure 3 nutrients-14-02213-f003:**
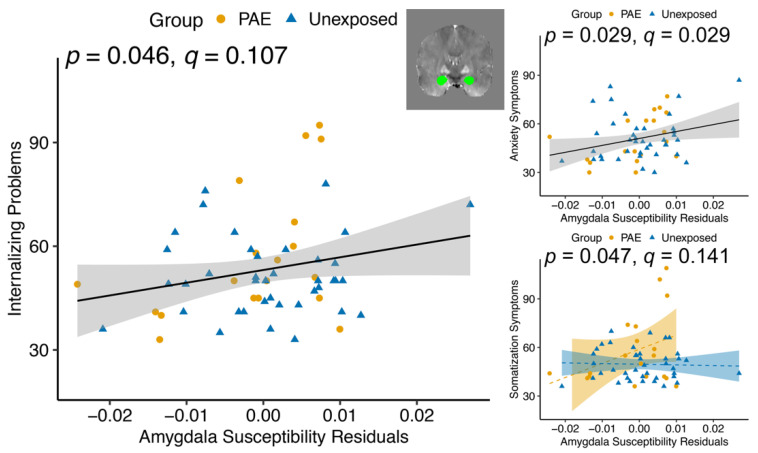
Amygdala magnetic susceptibility was positively associated with Internalizing Problems and Anxiety symptoms in youth with and without PAE. Group–susceptibility interactions showed higher susceptibility was associated with more Somatization symptoms in youth with PAE and fewer Somatization symptoms in unexposed youth, although associations were not significant within each group. Susceptibility values are corrected for age and gender. *p* values indicate the main effect of susceptibility on Internalizing Problems and Anxiety and the group–susceptibility interaction for Somatization. Dashed lines indicate non-significant associations within a group.

**Figure 4 nutrients-14-02213-f004:**
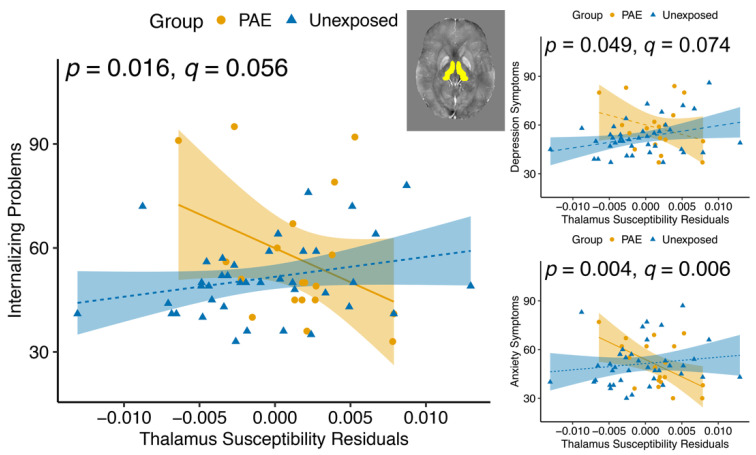
Group–susceptibility interactions in the thalamus were significant for Internalizing Problems, Depression, and Anxiety symptoms. Within the PAE group, susceptibility was more negatively associated with Internalizing Problems, Anxiety, and Depression symptoms than in unexposed controls. Susceptibility values are corrected for age and gender. *p* values indicate the group–susceptibility association. Dashed lines indicate non-significant associations within a group.

**Figure 5 nutrients-14-02213-f005:**
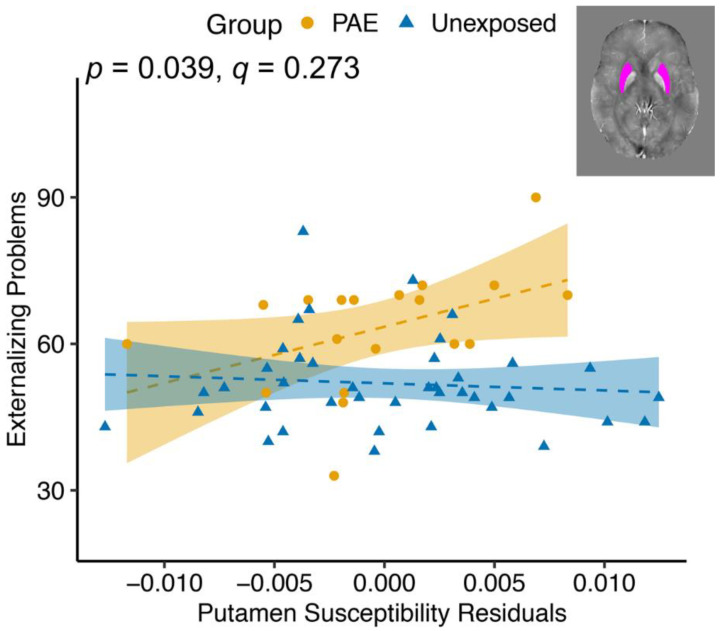
Group–susceptibility interactions in the putamen show that susceptibility is more positively associated with Externalizing Problems in youth with PAE compared to unexposed youth, though relationships were not significant within each group. Susceptibility values are corrected for age and gender. *p* value indicates the group–susceptibility association. Dashed lines indicate non-significant associations within a group.

**Table 1 nutrients-14-02213-t001:** Demographic information.

	Prenatal Alcohol Exposure (PAE)	Unexposed	Statistical Analysis
Age ± SD	11.18 ± 2.16	11.18 ± 2.27	*t* (37.057) = −0.005, *p* = 0.996
Parent/caregiver reported gender	53%B ^1^, 47%G (10B, 9G)	58%B, 43%G ^2^ (23B, 17G)	X^2^ (1) = 0.124, *p* = 0.725

^1^ B = boy, G = girl. ^2^ Due to rounding, numbers may not sum to 100%.

**Table 2 nutrients-14-02213-t002:** Group differences in susceptibility between PAE and unexposed control participants by region of interest.

	Mean Susceptibility ± SD (ppm)			
Brain Region	PAE	Control	F	*p*(*q*)
Caudate	0.0236 ± 0.00565	0.0215 ± 0.00682	1.45	0.234 (0.546)
Putamen	0.0233 ± 0.00563	0.0240 ± 0.00634	0.114	0.737 (0.737)
Pallidum	0.0831 ± 0.0147	0.0816 ± 0.0182	0.175	0.677 (0.737)
Thalamus	0.00322 ± 0.00374	0.000945 ± 0.00519	2.71	0.105 (0.368)
Amygdala	0.0171 ± 0.00926	0.0186 ± 0.00971	0.367	0.547 (0.737)
Hippocampus	−0.00177 ± 0.00977	0.00340 ± 0.00814	4.57	0.037 * (0.259)
Nucleus accumbens	−0.0336 ± 0.0154	−0.0384 ± 0.0200	0.918	0.342 (0.598)

* *p*< 0.05.

**Table 3 nutrients-14-02213-t003:** Group differences in volume between PAE and unexposed control participants by region of interest.

	Mean Volume ± SD mm^3^			
Brain Region	PAE	Control	F	*p(q)*
Caudate	6801 ± 686	7592 ± 738	16.7	<0.001 ** (<0.001 **)
Putamen	9812 ± 955	10,548 ± 1128	5.92	0.018 * (0.042 *)
Pallidum	3189 ± 356	3639 ± 401	18.2	<0.001 ** (<0.001 **)
Thalamus	15,227± 1707	15,830 ± 1442	1.963	0.167 (0.234)
Amygdala	3143 ± 338	3281 ± 396	1.68	0.201 (0.235)
Hippocampus	7912 ± 915	8277 ± 652	2.90	0.094 (0.164)
Nucleus accumbens	1212 ± 214	1239 ± 192	0.180	0.673 (0.673)

* *p*< 0.05, ** *p* < 0.001.

**Table 4 nutrients-14-02213-t004:** Comparison of Externalizing and Internalizing BASC-2-PRS *T* scores by diagnostic group.

		Diagnostic Group		Wilcoxon Rank Test
		PAE (*n* = 19)	Unexposed Controls (*n* = 40)	
Externalizing Problems composite score	Median score (IQR) ^1^	68 (10)	50 (9)	*W* = 604, *p* < 0.001 **
	At risk (60–69)	9 (47%)	4 (10%)	
	Clinically significant ≥ 70	5 (26%)	2 (5%)	
	Total	14 (73%)	6 (15%)	
Aggression subscale	Median score (IQR)	60 (11)	50 (8)	*W* = 604, *p* < 0.001 **
	At risk (60–69)	8 (42%)	2 (5%)	
	Clinically significant ≥ 70	2 (11%)	2 (5%)	
	Total	10 (53%)	4 (10%)	
Conduct subscale	Median score (IQR)	60 (16)	51 (14)	*W* = 575, *p* = 0.002 *
	At risk (60–69)	5 (26%)	5 (12.5%)	
	Clinically significant ≥ 70	5 (26%)	3 (7.5%)	
	Total	10 (52%)	8 (20%)	
Hyperactivity subscale	Median score (IQR)	69 (14)	50 (10)	*W* = 626, *p* < 0.001 **
	At risk (60–69)	8 (42%)	5 (12.5%)	
	Clinically significant ≥ 70	6 (32%)	2 (5%)	
	Total	14 (74%)	8 (17.5%)	
Internalizing Problems composite score	Median score (IQR)	50 (19)	50 (15)	*W* = 423, *p* = 0.490
	At risk (60–69)	2 (11%)	3 (7.5%)	
	Clinically significant ≥ 70	4 (21%)	4 (10%)	
	Total	6 (32%)	7 (17.5%)	
Anxiety subscale	Median score (IQR)	49 (23)	49 (16)	*W* = 372, *p* = 0.900
	At risk (60–69)	5 (26%)	3 (8%)	
	Clinically significant ≥ 70	2 (11%)	5 (13%)	
	Total	7 (37%)	8 (20%)	
Depression subscale	Median score (IQR)	58 (15)	51 (11)	*W* = 477, *p* = 0.119
	At risk (60–69)	4 (21%)	3 (8%)	
	Clinically significant ≥ 70	4 (21%)	4 (10%)	
	Total	8 (42%)	7 (18%)	
Somatization subscale	Median score (IQR)	50 (27)	48 (14)	*W* = 428, *p* = 0.445
	At risk (60–69)	1 (5%)	6 (15%)	
	Clinically significant ≥ 70	5 (26%)	1 (3%)	
	Total	6 (32%)	7 (18%)	

^1^ IQR = interquartile range, * *p* < 0.05, ** *p* < 0.001.

**Table 5 nutrients-14-02213-t005:** Associations between susceptibility and mental health symptoms in brain regions. Internalizing and Externalizing subscales were only assessed in brain regions with a significant main effect of susceptibility or a group–susceptibility interaction for Internalizing or Externalizing composite scores.

	Brain Region	Main Effect of Diagnostic Group		Main Effect of Susceptibility		Group–Susceptibility Interaction ^1^	
		F	*p*	F	*p*	F	*p*
Internalizing Problems composite score	Caudate	1.66	0.203	0.213	0.646		
	Putamen	1.87	0.177	0.048	0.828		
	Pallidum	1.73	0.194	3.28	0.076		
	Thalamus	1.81	0.185	0.091	0.764	6.16	0.016 *
	Amygdala	2.54	0.117	4.17	0.046 *		
	Hippocampus	2.39	0.128	0.598	0.443		
	Nucleus accumbens	3.21	0.079	7.02	0.011 *		
Anxiety subscale	Thalamus	0.024	0.879	0.043	0.837	9.31	0.004 *
	Amygdala	.000215	0.988	5.01	0.029 *		
	Nucleus accumbens	0.060	0.808	12.46	<0.001 **		
Depression subscale	Thalamus	2.24	0.140	1.10	0.300	4.04	0.049 *
	Nucleus accumbens	4.29	0.043 *	5.45	0.023 *		
Somatization subscale	Amygdala	4.54	0.038 *	1.52	0.222	4.12	0.047 *
Externalizing Problems composite score	Caudate	13.5	<0.001 **	0.911	0.344		
	Putamen	16.1	<0.001 **	0.432	0.514	4.50	0.039 *
	Pallidum	14.6	<0.001 **	0.225	0.637		
	Thalamus	12.2	<0.001 **	2.64	0.110		
	Amygdala	14.4	<0.001 **	0.471	0.495		
	Hippocampus	14.2	<0.001 **	0.061	0.806		
	Nucleus accumbens	14.4	<0.001 **	0.014	0.907		
Aggression subscale	Putamen	14.8	<0.001 **	0.332	0.567		
Conduct subscale	Putamen	12.0	0.001 *	3.57	0.062		
Hyperactivity subscale	Putamen	20.2	<0.001 **	0.883	0.352		

^1^ When the group–susceptibility interaction was not significant, it was removed from the final model, * *p* < 0.05, ** *p* < 0.001.

**Table 6 nutrients-14-02213-t006:** Associations between volume and mental health symptoms in brain regions.

	Brain Region	Main Effect of Diagnostic Group		Main Effect of Volume	
		F	*p*	F	*p*
Internalizing Problems composite score	Caudate	1.90	0.174	0.124	0.728
	Putamen	2.19	0.145	0.282	0.598
	Pallidum	1.46	0.232	0.000713	0.979
	Thalamus	1.55	0.218	0.370	0.546
	Amygdala	1.90	0.174	0.012	0.912
	Hippocampus	2.12	0.152	0.233	0.631
	Nucleus accumbens	2.33	0.132	3.52	0.066
Externalizing Problems composite score	Caudate	12.4	<0.001 **	0.102	0.750
	Putamen	11.6	0.001 **	0.760	0.387
	Pallidum	6.04	0.017 *	3.99	0.051
	Thalamus	12.8	<0.001 **	2.39	0.128
	Amygdala	14.4	<0.001 **	0.003	0.955
	Hippocampus	13.7	<0.001 **	0.039	0.844
	Nucleus accumbens	15.6	<0.001 **	1.32	0.256

* *p* < 0.05, ** *p* < 0.001.

## Data Availability

Unexposed participant T1-weighted neuroimaging data are available at: https://figshare.com/articles/dataset/Structural_and_diffusion-weighted_images_of_the_adolescent_brain/6002273. PAE data are available upon request to the principal investigator.

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
