# Peer review of "Brain Iron and Mental Health Symptoms in Youth with and without Prenatal Alcohol Exposure"

_nutrients, 2022, doi:10.3390/nu14112213_

Round 1
Reviewer 1 Report
This is a potentially important paper that extends current clinical and preclinical studies and reports an association between the magnetic susceptibility of select brain regions – a surrogate for iron content – and externalizing and internalizing behaviors in people who experienced PAE. A growing literature increasingly suggests that iron status is a significant modulator of outcomes in response to PAE. The current manuscript adds to this evidence base through identifying associations between brain iron and behavior. The methodology is validated in prior clinical and post-mortem studies, and is a potentially elegant method to address the study’s hypothesis. There are several limitations, addressable by the authors, that will enhance the study’s value.
Importantly, the authors’ need to explain what their MRI method is actually detecting, and then to interpret its implications. First, please add a discussion – probably to the Discussion – on what iron forms are being detected by this method. Presumably, these are the trace quantities of Iron-57 plus the trace quantities of magnetite that are known to be present in human brain tissue (i.e., PMC1885366); I doubt the method can distinguish these. The former would include heme iron, FeS clusters, and storage iron (ferritin) as the dominant forms.
Then, in light of this information, the Discussion needs a clearer interpretation on what the findings suggest. The current discussion on this point is overly simplistic; for example, it does not acknowledge that the method cannot distinguish between bioavailable and storage iron. There are suggestions that PAE alters the brain vasculature, and such differences could impact the contribution of vascular heme iron from the red cells to the susceptibility measure.
The discussion also ignores the nuance that subregion iron content does not reflect need. Brain subregions differ greatly in their iron content and some with a low content (i.e. hippocampus) have a high need. Conversely, high iron content may reflect a storage capacity. This is relevant to the discussion of negative and positive correlations.
How were the composite scores calculated? This needs more explanation as it is a main measure in the analyses. Similarly, seven brain regions and eight behaviors (including the composites) were evaluated. Please provide data for all the analyses including those that lacked an association; this could be done in a table. Negative data are also important!
While the role of iron status on dopaminergic function is well-studied, this is hardly iron’s only role in neurotransmission. All or nearly all of the synthetic and catabolic enzymes mediating neurotransmitter content are iron-dependent. This broader actions for iron need to be acknowledged.
Minor comments:
Justification for the select brain regions is unclear.
Data on hippocampus were not presented or discussed, which is odd as it highly vulnerable to iron status.
Why do the figures only present the p values and not the adjusted q values? Some associations significant in the p lost significance in the FDR.
Provide the participant characteristics in a new Table 1, rather than buried in the Methods.
MW Miller and colleagues (PMID 7790882) were the first to demonstrate that PAE alters the iron content of brain subregions, and this work needs to be cited and acknowledged.
Reviewer 2 Report
Overall, this is a novel and interesting paper. The biggest concern is presentation of some non-significant data as significant.
Abstract – clarify what is meant by “susceptibility”. For example “We investigated if brain susceptibility, an indirect measurement of brain iron,…”
Line 98-101 – what is the “previous literature” referred to? The statement has no references.
Lines 114 & 130 – are the first numbers in the parentheses reporting mean age? And is this reporting standard deviation or standard error?
Section 3.2 Behavioral outcomes – this section refers to median scores but the tables show mean scores. Externalizing behaviors are actually in Table 2, not 1 as referenced on line 207. The table for internalizing behaviors should be referenced in the text.
Axis labels on the smaller graphs should be enlarged so they can be read.
Please check q value on line 224.
Figure 3 – the legend (lines 231-234) suggests that there’s a significant positive association between susceptibility and somatization in PAE but the line is dashed and the statement on lines 235-236 notes that dashed lines indicate non-significant associations. If associations for both subgroups are non-significant that should be clear in the legend text. Please clarify what association the p value is for.
Figures 4 & 5 – legends. As above, please clarify what is significant and what is not, and what association the p value is for.
Why are the data for brain volumes and QSM not included? Were there any differences between the groups in these measures?
In the Discussion the non-significant associations for PAE or Unexposed are treated as significant. If the authors define what they consider a trend they could report those outcomes (as trends) throughout the paper, but it must be clear what is significant and what is not.
Round 2
Reviewer 1 Report
The manuscript is substantially and thoughtfully improved, and represents an important contribution to our understanding of how PAE impacts brain iron. We appreciate your adding the additional information, which adds to the m/s relevance and provides context for the data. The authors have satisfactorily addressed concerns, save two minor comments:
Line 337: thank you for clarifying the iron forms detected here. Ferritin is the primary iron storage form. If it is increased in PAE, you might consider whether this iron is available for use, or may instead be unavailable, as per refs 36 & 37. Alternately, it might reflect an overload situation.
Line 418-419: "Animal research has been inconclusive whether iron supplementation can reverse 418 the effects of iron deficiency during development [38,93]," This is a misinterpretation of the data. The data are not inconclusive. They show that select features of PAE can be normalized by supplemental iron. Also see Helfrich KK in this issue (doi: 10.3390/nu14081653.) It is not unexpected that any single nutrient (or intervention) would fix everything, given there is no single 'alcohol receptor' and thus is actions are pleiotropic.
Author Response
The manuscript is substantially and thoughtfully improved, and represents an important contribution to our understanding of how PAE impacts brain iron. We appreciate your adding the additional information, which adds to the m/s relevance and provides context for the data. The authors have satisfactorily addressed concerns, save two minor comments:
1. Line 337: thank you for clarifying the iron forms detected here. Ferritin is the primary iron storage form. If it is increased in PAE, you might consider whether this iron is available for use, or may instead be unavailable, as per refs 36 & 37. Alternately, it might reflect an overload situation.
Thank you for this thoughtful comment. The following has been added on line 339:
Increased ferritin iron stores may reflect sufficient brain iron available for use, stored iron that is unavailable, or an overload of brain iron.
2. Line 418-419: "Animal research has been inconclusive whether iron supplementation can reverse 418 the effects of iron deficiency during development [38,93]," This is a misinterpretation of the data. The data are not inconclusive. They show that select features of PAE can be normalized by supplemental iron. Also see Helfrich KK in this issue (doi: 10.3390/nu14081653.) It is not unexpected that any single nutrient (or intervention) would fix everything, given there is no single 'alcohol receptor' and thus is actions are pleiotropic.
The following has been edited on line 425:
Animal research has shown iron supplementation can mitigate some of the effects of alcohol exposure during development [38,93-94].